# In Vitro Evaluation of Antiviral Activities of Funicone-like Compounds Vermistatin and Penisimplicissin against Canine Coronavirus Infection

**DOI:** 10.3390/antibiotics12081319

**Published:** 2023-08-15

**Authors:** Claudia Cerracchio, Maria Michela Salvatore, Luca Del Sorbo, Francesco Serra, Maria Grazia Amoroso, Marina DellaGreca, Rosario Nicoletti, Anna Andolfi, Filomena Fiorito

**Affiliations:** 1Department of Veterinary Medicine and Animal Production, University of Naples Federico II, 80137 Naples, Italy; claudia.cerracchio@unina.it (C.C.); luca.delsorbo2@studenti.unina.it (L.D.S.); 2Department of Chemical Sciences, University of Naples Federico II, 80126 Naples, Italy; mariamichela.salvatore@unina.it (M.M.S.); dellagre@unina.it (M.D.); 3Institute for Sustainable Plant Protection, National Research Council, 80055 Portici, Italy; 4Istituto Zooprofilattico Sperimentale del Mezzogiorno, Unit of Virology, Department of Animal Health, 80055 Portici, Italy; francesco.serra@izsmportici.it; 5Department of Agricultural Sciences, University of Naples Federico II, 80055 Portici, Italy; rosario.nicoletti@crea.gov.it; 6Council for Agricultural Research and Economics, Research Centre for Olive, Fruit and Citrus Crops, 81100 Caserta, Italy; 7BAT Center-Interuniversity Center for Studies on Bioinspired Agro-Environmental Technology, University of Naples Federico II, 80055 Portici, Italy

**Keywords:** secondary fungal metabolites, *Talaromyces pinophilus*, canine coronavirus, A72 cells, virus yield, viral nucleocapsid protein, aryl hydrocarbon receptor, lysosome acidity

## Abstract

Recent studies have demonstrated that 3-*O*-methylfunicone (OMF), a fungal secondary metabolite from *Talaromyces pinophilus* belonging to the class of funicone-like compounds, has antiviral activity against canine coronaviruses (CCoV), which causes enteritis in dogs. Herein, we selected two additional funicone-like compounds named vermistatin (VER) and penisimplicissin (PS) and investigated their inhibitory activity towards CCoV infection. Thus, both compounds have been tested for their cytotoxicity and for antiviral activity against CCoV in A72 cells, a fibrosarcoma cell line suitable for investigating CCoV. Our findings showed an increase in cell viability, with an improvement of morphological features in CCoV-infected cells at the non-toxic doses of 1 μM for VER and 0.5 μM for PS. In addition, we observed that these compounds caused a strong inhibition in the expression of the aryl hydrocarbon receptor (AhR), a ligand-activated transcription factor which is activated during CCoV infection. Our results also showed the alkalinization of lysosomes in the presence of VER or PS, which may be involved in the observed antiviral activities.

## 1. Introduction

Fungal secondary metabolites have been recognized for many years as a major source for drug discovery due to the wide range of diversity of their chemical structures and biological activities [1,2,3,4,5,6]. Fungi of the genus *Talaromyces* are a store house of bioactive compounds with huge structural diversity belonging to diverse classes of natural products, such as alkaloids, anthraquinones, benzofurans and isocoumarins [7,8,9,10]. Due to their remarkable bioactivities, these compounds could play a crucial role in drug development against various diseases.

Funicone-like compounds represent target metabolites of *Talaromyces* species because they are frequently produced by fungi from this genus. This homogeneous group of secondary metabolites is characterized by a molecular structure based on a γ-pyrone ring linked through a ketone group to an α-resorcylic acid nucleus [11]. Many of these compounds have been documented as potential drug candidates due to their valuable bioactivities, including antibiotic, antiviral, cytotoxic and insecticidal properties [12,13,14,15,16,17].

Recently, global interest in the antiviral properties of fungal secondary metabolites has increased due to the novel severe acute respiratory syndrome coronavirus (SARS-CoV-2). Owing to their high chemical diversity, secondary metabolites from *Talaromyces* species offer great promise as potentially effective antiviral drugs.

Within our recent activity aimed at studying the antiviral properties of funicone-like compounds, the antiviral activities of 3-*O*-methylfunicone (OMF), a member of this group of secondary metabolites obtained from *Talaromyces pinophilus,* have been investigated. Our results showed that OMF is able to reduce the infectivity of bovine herpesvirus-1 [18] and canine coronavirus (CCoV) [19].

Hence, these promising results encouraged us to conduct further research on antiviral properties of two additional funicone-like compounds named penisimplicissin (PS) and vermistatin (VER). The latter represents the most common funicone-like compound, having been reported as a product of at least 15 fungal species [7,9,11]. VER and PS have a very similar molecular structure (Figure 1) which is characterized by a 4,6-dimethoxyphthalide moiety linked to a γ-pyridone [11].

In this study, the potential antiviral effects of VER and PS on CCoV infection in canine fibrosarcoma cells (A72) have been investigated. Our interest in CCoV derives from the urgent need to find effective preventive and therapeutic measures to develop antiviral treatments. In fact, since 2005, pantropic strains of this virus have emerged, provoking lethal multi-systemic disease especially in young puppies [20,21,22,23,24]. Subsequently, in different countries, harmful novel CCoV strains, such as CCoV-HuPn-2018 [25] and HuCCoV_Z19Haiti [26], emerged from zoonotic spillover events from wild animals into humans, causing acute respiratory illness. Following gene recombination and mutation, new strains of CCoV can emerge and spread, after transmission from one reservoir species to new host species [27,28].

## 2. Results

### 2.1. Funicone-like Compounds VER and PS Have Antiviral Activity against CCoV Infection

Vermistatin (VER) and penisimplicissin (PS) (Figure 1) were purified from mycelium of isolate LT6 of *T. pinophilus* and identified on the basis of their spectroscopic data [29].

To investigate the effect of VER and PS in CCoV infection, cell viability was assessed by the trypan blue (TB) exclusion test. We developed a dose-response curve after exposure of A72 cells to different doses of VER and PS (Figure 2A–D). VER at 0.1, 0.5 and 1 µM, as well as PS at 0.1, 0.5 and 2.5 µM, induced no differences in cell viability (*p* > 0.05) (Figure 2C,D). This non-monotonic dose-response was previously observed in A72 and bovine (MBDK) cells treated with OMF [18,19]. In addition, in A72 cells, IC_50_ was obtained with 4.2556 μM VER and 4.9562 PS μM, respectively.

We subsequently detected cell growth inhibition in CCoV infected cells treated with VER and PS (Figure 3A–D). Following CCoV infection, both VER and PS significantly increased cell viability (*p* < 0.01 and *p* < 0.05) of A72 cells (Figure 3B,C). Thus, VER at 1 µM and PS at 0.5 µM were chosen to be used throughout the study.

Analysis of cell morphology was carried out to assess the effect of both VER and PS during CCoV infection. After Giemsa staining, light microscopy analysis was performed. In Figure 4A, the comparison between CCoV-infected cells treated or not with VER and PS displayed changes in cell morphology. Features of cell death, such as cellular shrinkage (Figure 4A, arrowhead), pyknosis (Figure 4A, arrow), were noticeably lessened by both funicone-like compounds (Figure 4A). Furthermore, an enhancement of intercellular spaces provoked by the detachment of cells from the culture plate was mostly detected in untreated CCoV-infected cells (Figure 4A, slim arrow). After treating with acridine orange/propidium iodide (AO/PI), fluorochromes used for the detection of both viable and dead cells, cells were observed by fluorescence microscopy. AO, which is membrane-permeable, binds to nucleic acids, provoking a green fluorescence. PI, impermeable to intact cell membrane, crosses the membrane of dead and dying cells and intercalates with nucleic acids, forming a bright red fluorescent complex. The combination of both fluorescent probes allows the simultaneous detection of cells with intact or compromised cell membranes [30]. In the presence of VER and PS a decrease in PI fluorescent cells was observed in infected cells compared to CCoV untreated groups (Figure 4B).

Overall, our results demonstrated that in A72 cells funicone-like compounds, like VER and PS, significantly reduced cell death, as well as morphological cell death signs during CCoV infection.

To explore the antiviral effect of VER and PS against CCoV infection, virus yield was evaluated after 48 h of infection in A72 cells. A significant (*p* < 0.01) decrease in virus titer was revealed by Quantitative Real-time RT-PCR (RT-qPCR) in VER- and PS-treated cells (Figure 5A). Additionally, a decreased cytopathic effect (CPE) was appreciable in both VER- and PS-treated groups compared to untreated infected cells (Figure 5B). All those effects were accompanied by a significant decrease (*p* < 0.001 and *p* < 0.01) in the expression of the viral nucleocapsid protein (NP), as detected in VER- and PS-treated cells compared to CCoV-infected groups (see Section 2.2).

Our observations showed that both VER and PS induce antiviral effects during CCoV infection.

### 2.2. AhR and Lysosomes Are Involved in Anti-CCoV Activity of VER and PS

Based on findings in previous studies [18,19,31], to preliminarily explore the mechanism of action of VER and PS in CCoV infection, we investigated the expression of AhR, a ligand-activated transcription factor which is up-regulated by CCoV. In the presence of both funicone-like compounds, a significant reduction in the expression of AhR was detected, in both uninfected (Figure 6A,B) and CCoV-infected cells (Figure 7A,B).

Lysosomes are cellular organelles, enclosing a variety of enzymes, which represent the digestive system of the cell. An acidic environment, developed by a proton pump, typifies lysosomes. The effect of VER and PS treatment on lysosomes was analyzed by LysoRed staining, a method used to label lysosomes in live cells. DMSO control cells showed an entirely acidified structure (Figure 8). In contrast, deacidification was observed in the presence of both funicone-like compounds in a significant number of cells (Figure 8). During CCoV infection in A72, we observed cellular deacidification (Figure 9), which were further alkalinized by both funicone-like compounds (Figure 10).

## 3. Discussion

Coronaviruses (CoVs) are responsible for a variety of diseases ranging from bronchitis to gastroenteritis, infectious peritonitis, encephalitis and hepatitis in mammalians and birds [32]. Human CoVs are often originated from animals and then adapted to humans by direct jumping or by jumping into an intermediate animal host [33]. Before the outbreak in 2003 of severe acute respiratory syndrome (SARS)-CoV, which was the first highly pathogenic human CoV, there was little knowledge of coronaviruses and, in general, researchers focused their attention on animal CoVs.

One of the most studied animal CoVs is the canine coronavirus (CCoV) which is an enveloped, positive-stranded RNA virus belonging to the genus of alphacoronaviruses [28].

Recent investigations showed that the funicone-like compound 3-*O*-methylfunicone (OMF) is a promising antiviral agent against CCoV [19]. Funicone-like compounds are a homogeneous class of fungal secondary metabolites characterized by valuable bioactivities and have potential as drug candidates [11].

In this study, the funicone-like compounds vermistatin and penisimplicissin have shown antiviral properties against CCoV, and PS turned out to be more active than VER. In fact, our data show an increase in cell viability, with an improvement of morphological features in CCoV-infected cells, at the non-toxic doses of 1 μM for VER and at 0.5 μM for PS. In addition, a substantial decrease in virus yield was observed, accompanied by a reduction in the expression of viral protein NP. Similar promising antiviral results against CCoV were previously obtained using OMF [19].

Understanding the potential mechanism of action of funicones represents a fascinating challenge because of the involvement of a deeply mysterious actor, AhR. This receptor shows a regulatory activity of immune functions in response to endogenous metabolites (i.e., bilirubin, biliverdin and tryptophan) as well as exogenous ligands (i.e., dietary flavonoids, environmental contaminants and microbial metabolites) [34,35]. Increasing evidence underlines the role of AhR in CoV infection [36]. Indeed, both alphacoronavirus (H-CoV-229E and CCoV) and betacoronavirus (MCoV, MERS-CoV, SARS CoV-1 and SARS-CoV-2) up-regulate AhR during infection in vitro [31,37,38,39,40,41], indicating AhR as a feasible target for antiviral therapy. Interestingly, the blocking of AhR, pharmacologically induced by CH223191 (or by OMF), reduces the replication of CCoV in vitro [19,31]. Here, the CCoV-induced activation of AhR was noticeably reduced by VER and PS during infection. These findings are further evidence of a blocking action of AhR induced by funicone-like compounds with a promising antiviral property.

An acidic environment characterizes lysosomes, due to a proton pump V-ATPase complex pumping H^+^ from the cytoplasm into the endo-lysosome. The V-ATPase complex represents a key factor for viral entry into the host cell [42,43]. In fact, the acidic lysosomal environment is required for lysosomal enzyme stability and activity, and even a small increase in pH is sufficient to inhibit these enzymes. Therefore, the pharmacological modulation of lysosomal pH can interfere with the endosomal pathway and intracellular membrane trafficking crucial for viral infection. Hence, lysosomotropic agents (e.g., chloroquine, hydroxychloroquine or azithromycin) are able to prevent CoV infection representing new therapeutic strategies. In fact, the basic amine property of chloroquine and similar molecules leads to their accumulation in cellular acidic compartments and raises their pH [44,45,46,47,48].

However, a significant deacidification of lysosomes during CoV infection was reported, along with a reduction in lysosomal enzyme activity. This deacidification mechanism is currently under investigation, but it was hypothesized that lysosomes become deacidified indirectly due to an excessive cargo (i.e., viruses) and/or perturbations in the proton pump or ion channel trafficking [44].

In this work we observed that the acidic environment of A72 cells was deacidified by CCoV infection and this is in agreement with what is described above regarding CoV infection. In addition, at non-toxic doses both funicone-like compounds tested were active on cells, either CCoV-infected or not, inducing alkalinization of lysosomes. However, VER or PS could be not directly responsible for the observed increase in lysosome pH because, unlike the above-mentioned lysomotropic agents, these compounds do not have basic properties. For this reason, it can be deduced that the alkalinization of the acidic lysosomes is the result of a more complex interaction between funicone-like compounds and lysosomes. In fact, several molecular mechanisms could be implicated in the alteration of the lysosome functions such as the ones reported for diverse mycotoxins. For instance, citrinin is responsible for lysosome damage in mouse oocytes with the increased expression of LAMP2 (lysosomal associated protein 2) [49], a lysosomal marker protein which protects the lysosomal membrane from autodigestion and maintains the lysosomal acidic environment [50]. Zearalenone was reported to upregulate the expression of LAMP2, LC3 (microtubule-associated protein 1 light chain 3 alpha) and ATG7 (autophagy-related 7) in porcine oocytes [51]. Sterigmatocystin led to lysosomal leakage [52], while citreoviridin exposure caused lysosomal dysfunction in human liver-derived HepG2 cells [53]. Exposure to fumonisin B1 in bovine kidney cells and human intestinal Caco-2 cells impaired lysosome function by affecting lysosome integrity [54].

We conjecture that the observed differences in the antiviral activities of VER and PS could be related to their functionalization. In fact, PS shows a methyl group while VER shows a propenyl group linked to the γ-pyrone ring (Figure 1). This means that this molecular site highly contributes to their antiviral activities. However, additional investigations of the antiviral properties of unexplored funicone-like compounds may provide better insights into their structure-activity relationship.

## 4. Materials and Methods

### 4.1. Production and Isolation of Funicone-like Compounds

Cultures of *T. pinophilus* (strain LT6) were prepared as previously reported [29]. The fresh mycelium obtained from 6 L of liquid cultures after filtration was homogenized in a Waring blender 7011HS (Waring, Torrington, CT, USA) for 5 min at high speed with 440 mL of MeOH-H_2_O (NaCl 1%) mixture (55:45 *v/v*). The supernatant was separated by centrifugation (30 min at 7000 rpm, 10 °C), and the residue was homogenized with 250 mL of the initial hydroalcoholic solution. The resulting suspension was separated as reported above. The two supernatants were combined and extracted with CHCl_3_ (600 mL, three times). The organic phase was anhydrified with Na_2_SO_4_ and evaporated under reduced pressure. The obtained crude extract (230.2 mg) was purified through a chromatographic column (1.5 × 40 cm i.d.) on silica gel (Kieselgel 60, 0.063–0.200 mm, Merk, Darmstadt, Germany), eluted with CHCl_3_/*iso*-PrOH (97:3 *v/v*), yielding seven homogeneous fractions. The residue of the third fraction was purified by TLC on silica gel (Kieselgel 60, F254, 0.25 mm, Merk) eluted with *n*-hexane/acetone (6:4, *v/v*) to afford a crystalline compound identified as vermistatin, and an amorphous solid identified as penisimplicissin (1.5 and 0.5 mg, R_f_ 0.37 and 0.29 on TLC in the same chromatographic conditions, respectively). They were identified by comparing the NMR data with the ones of a previous report [29].

### 4.2. Cell Cultures and Virus Infection

A72 cells (ATCC, CRL-1542) [55], derived from canine fibrosarcoma cell line, were cultured in Dulbecco’s modified Eagle’s minimal essential medium (DMEM) at 37 °C and 5% CO_2_ [56,57]. CCoV type II (strain S/378, GenBank accession number KC175341) was used throughout the study. A72 cells were used for virus stocks growth as well as for virus titration [56].

VER and PS were dissolved in DMSO, then added to the medium to obtain final concentrations of 0.1, 0.5, 1, 2.5 and 5 μM. Vehicle control was DMSO diluted in DMEM (0.1% *v/v*).

Cells, in monolayers, were infected with CCoV, at different multiplicity of infection (MOI), exposed to VER or PS, to produce four groups: CCoV uninfected or infected cells, VER- or PS-treated infected and uninfected cells. After adsorption at 37 °C (one hour), A72 cells were incubated, and processed at 24 and 48 h post infection. The virus remained in the medium.

### 4.3. Cell Viability

TB (Sigma-Aldrich, St. Louis, MO, USA) exclusion test was used to assess cell viability [18]. Monolayers of cells were infected with CCoV at MOI of 0.05 and treated with VER or PS incubated for 48 h, then analyzed as reported [18]. Cell viability was calculated as % of viable cells over total cell number, and results are the mean ± S.D. from three separate experiments with duplicated samples. In order to calculate the IC_50_ in A72 cells, IC50 Calculator|AAT Bioquest (https://www.aatbio.com/tools/ic50-calculator accessed on 10 August 2023) was employed.

### 4.4. Examination of Cell Morphology

Giemsa staining and acridine orange/propidium iodide (AO/PI) were used to analyze cell morphology [30,58,59]. A72 cells were infected with CCoV, at MOI of 1, and treated or not with VER or PS. After 24 h of infection, Giemsa staining and AO/PI were performed. Light microscopy examination was then performed using a ZOE Cell Imager (Bio-Rad Laboratories, Hercules, CA, USA). The identification of cell death features were assessed by reported criteria [60,61,62]. In addition, living and dead cells were simultaneously identified as a previous study reported [30].

### 4.5. LysoRed Staining

The cells were infected with CCoV, at MOI of 5, treated with VER or PS, and incubated for 24 and 48 h. In accordance with the user manual, cells were stained with CytoPainter LysoRed Indicator Reagent (Abcam, Waltham, MA, USA) and incubated. The cells were then washed and observed under a fluorescence microscope [63].

### 4.6. Immunofluorescence (IF) Staining

A72 cells were infected with CCoV at MOI 5 and treated with VER or PS for 24 h. IF staining was then carried out according to previously published protocols [19,31,64]. The following antibodies, dissolved in 5% bovine serum albumin-TBST, were used: anti-NP monoclonal mouse, MAB 938 (The Native Antigen Company, Kidlington, UK), anti-AhR (Sigma-Aldrich, St. Louis, MO, USA) (1:250), Texas Red goat anti-rabbit (Thermo Fisher Scientific, Waltham, MA, USA) (1:100) and Alexa Fluor 488 goat anti-mouse (Thermo Fisher Scientific) (1:1000). Microscopy was performed using a ZOE Fluorescent Cell Imager (Bio-Rad Laboratories) and quantification of fluorescence signals in selected images was assessed using ImageJ (National Institutes of Health) software (version 1.53a, National Institutes of Health, Bethesda, MD, USA).

### 4.7. Virus Production

Cells were infected with CCoV at MOI 5, treated with VER or PS and incubated for 24 and 48 h. Quantitative Real-time RT-PCR (RT-qPCR) was then used to quantify CCoV. Additionally, CPE was assessed by analyzing cells through light microscopy as previously reported [19,31].

### 4.8. Viral Nucleic Acids Extraction Procedures

Cell supernatant samples (200 µL) underwent nucleic acids extraction using an automatic extraction system (King Fisher Flex (Thermo Fisher Scientific, Waltham, MA, USA) with the Mag Max Viral Pathogen kit (Thermo Fisher Scientific, Waltham, MA, USA)) and following the manufacturer’s instructions. DMEM was employed as negative process control (NPC). Prior to extraction, an external process control (EPC), namely murine norovirus (10 µL) [65] (MNV-IT1 Acc. no. KR349276) (final stock concentration of 10^7^ PFU mL) was added to each sample (including NPC) in order to evaluate the likely presence of PCR inhibitors. EPC was amplified in each sample by real-time PCR with the following primers: MNoV F 50-CACGCCACCGATCTGTTCTG-30 and 50-GCGCTGCGCCATCACTC-30; and probe FAM-CGCTTTGGAACAATG-MGB-NFQ with the thermal profile indicated in the literature [66]. Results were analyzed as follows: if the threshold cycle (Ct) of the EPC in the eluted sample was comparable to that of the EPC in the NPC, the sample was analyzed as undiluted. If, instead, the difference between the two Cts was at least 3 or a multiple of 3, all the analyses were carried out on the sample diluted 1:10 or more (one decimal dilution for every 3 threshold cycles of difference) [67].

### 4.9. Real-Time RT-PCR for CCoV Quantification

CCoV viral load in A72 cells treated/not treated with funicone-like compounds was calculated in each sample by RT-qPCR using a standard curve (SC). Quantification was carried out at 1, 12, 24, 48 and 72 h post infection on a QuantStudio 5 Real-Time PCR instrument (Thermo Fisher Scientific) with the AgPath-ID™ One-Step RT-PCR Kit (Applied Biosystems–Thermo Fisher Scientific). The reaction (25 µL) contained: 5 µL extracted RNA, 12.5 µL reaction mixture, 1 µL reverse transcriptase-PCR enzyme mixture, 1 µL of primer CCoV-For (10 µM) (5′-TTGATCGTTTTTATAACGGTTC-TACAA-3′), 1 µL of primer CCoV-Rev (10 µM) (5′-AATGGGCCATAATAGCCACATAAT-3′) and 1 µL of probe CCoV-P (6 µM) (FAM-5′-ACCTCAATTTAGCTGGTTCGTGTATGGCATT-3′-TAMRA) [68]. The amplification was carried out employing the following thermal profile: one retrotranscription cycle of 30 min at 42 °C, followed by 15 min at 95 °C and 40 cycles of 15 s at 95 °C and 60 s at 60 °C. Preliminary experiments were carried out to set up the SC for viral quantification. To construct it, serial dilutions of the extracted virus (from 3.5 × 10^9^ to 3.5 × 10^4^ TCID_50_/mL) were analyzed in triplicate. The SC was then constructed by plotting the Log TCID_50_/mL versus the average Ct number obtained for each dilution. The resulting equation was y = 2.85x + 51.85 (E = 124,344%, r^2^ = 0.983). The SC was used in RT-qPCR assays [18,19,31] to estimate samples virus titer starting from the C_t_ obtained in the amplification reaction.

### 4.10. Statistical Analysis

Data are displayed as mean ± S.D. To assess one-way ANOVA with Tukey’s post-test, GraphPad InStat Version 3.00 for Windows 95 (GraphPad Software, San Diego, CA, USA) was utilized. A *p* value < 0.05 was deemed as statistically significant.

## Figures and Tables

**Figure 1 antibiotics-12-01319-f001:**
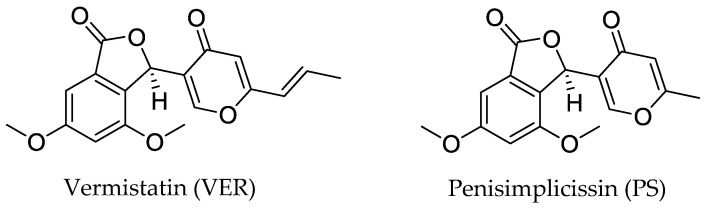
Chemical structures of vermistatin (VER) and penisimplicissin (PS).

**Figure 2 antibiotics-12-01319-f002:**
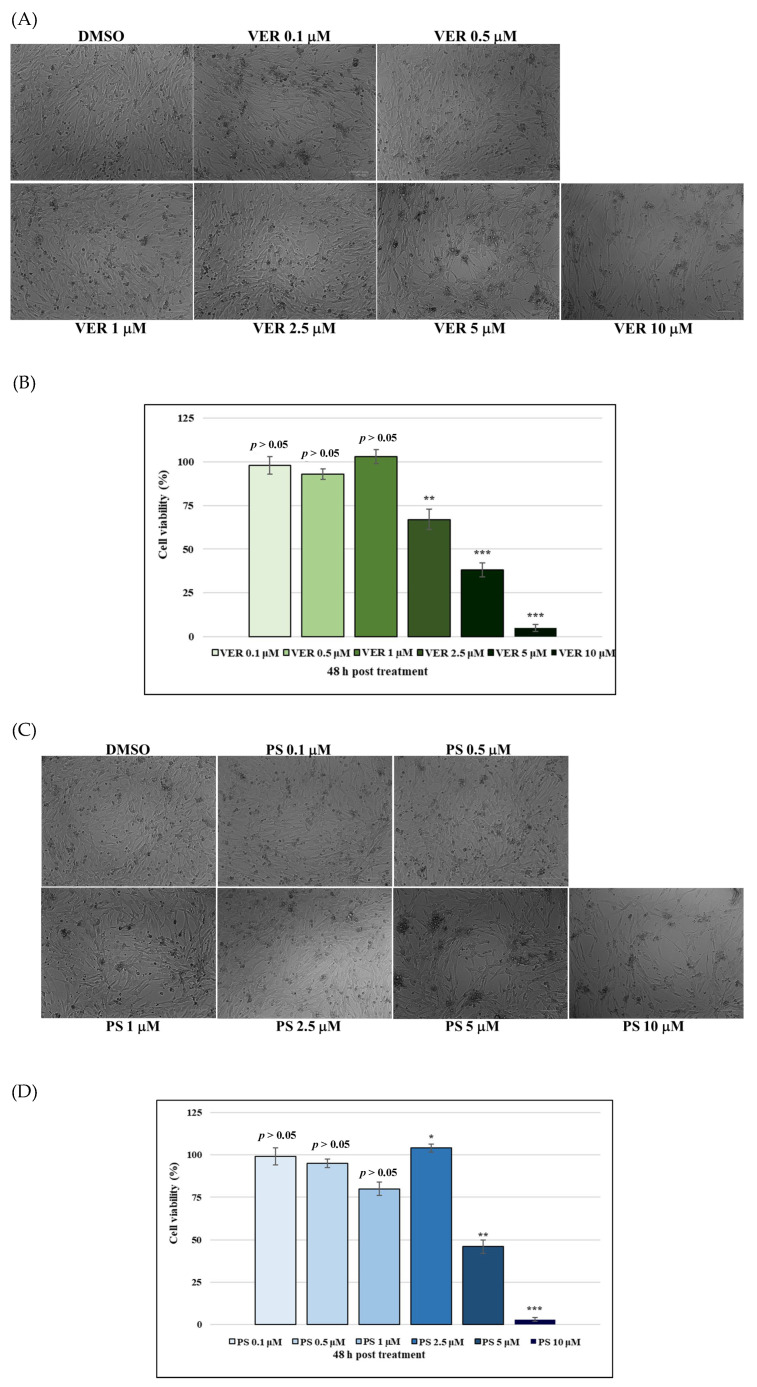
Identification of IC_50_ of VER and PS at various doses and development of dose-response curve. (**A**,**B**) Cells exposed to dimethyl sulfoxide (DMSO) or funicone-like compounds after 48 h of treatment. (**C**,**D**) Dose-response curve of A72 cells exposed to DMSO or both funicone-like compounds after 48 h of treatment. Scale bar 100 µm. Significant differences among DMSO and VER or PS groups were indicated by probability *p*. ** *p* < 0.01 and *** *p* < 0.001 for VER; * *p* < 0.05, ** *p* < 0.01 and *** *p* < 0.001 for PS.

**Figure 3 antibiotics-12-01319-f003:**
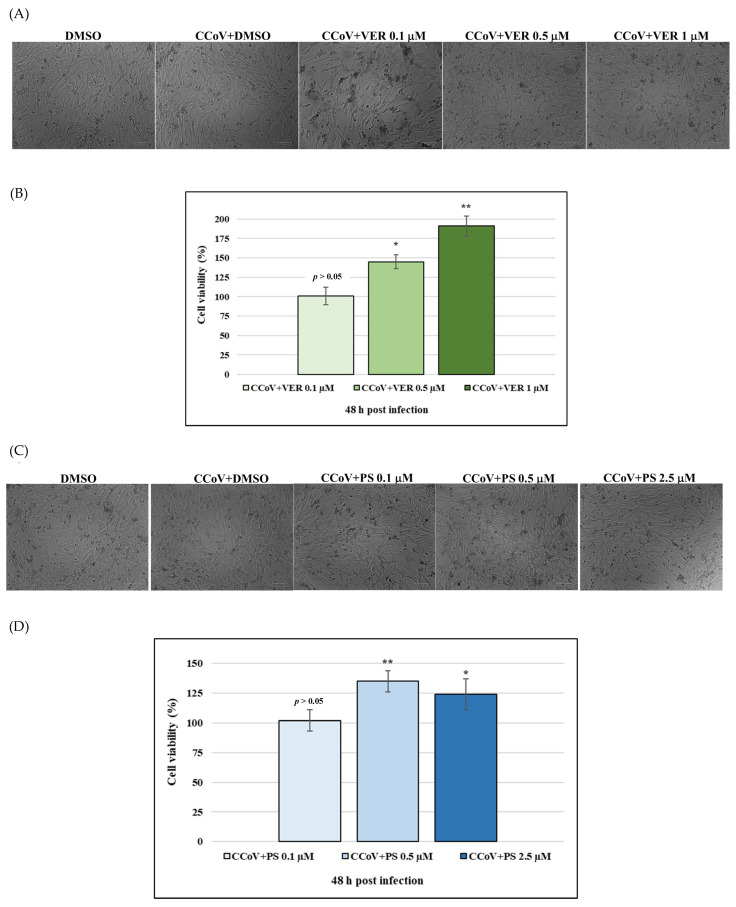
During CCoV infection VER and PS decrease cell death of A72 cells. (**A**,**B**) Cells infected with CCoV and treated with VER or PS at 48 h after infection (**C**,**D**) Dose-response curve of CCoV-infected cells treated with VER and PS for 48 h. Scale bar 100 µm. Significant differences among CCoV-infected and VER or PS-infected cells are indicated by probability *p*. ** *p* < 0.01 and * *p* < 0.05.

**Figure 4 antibiotics-12-01319-f004:**
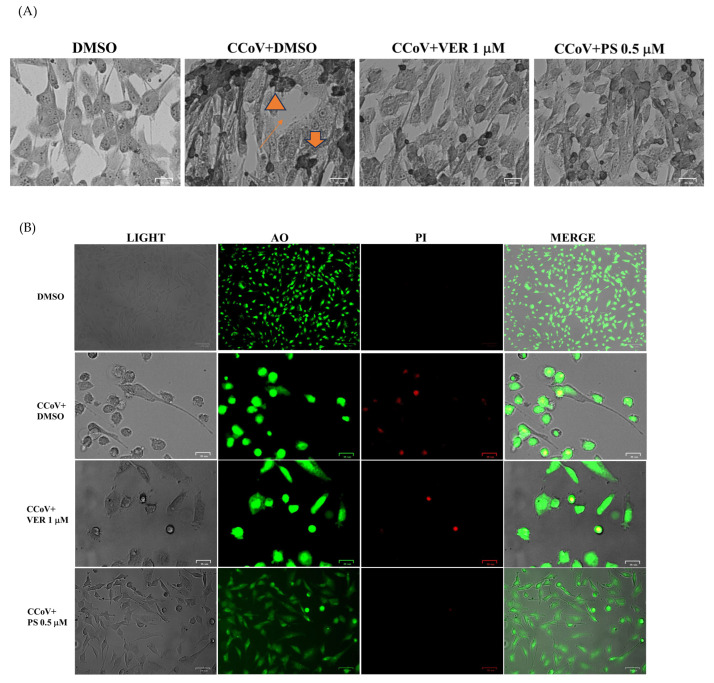
Funicone-like compounds reduce the typical morphological cell death features during CCoV infection. (**A**) Signs of cell death, such as cellular shrinkage (**A**, arrowhead), pyknosis (**A**, arrow), were noticeably lessened in VER- and PS-treated groups. Moreover, the enhancement of intercellular spaces produced by the detachment of cells from the culture plate was mainly observed in untreated CCoV-infected cells (**A**, slim arrow). (**B**) In AO/PI panels, PI fluorescent cells, indicating dead and dying cells, were mostly detected in CCoV-infected cells compared to CCoV-infected-cells treated with funicone-like compounds. Scale bar 25 µm and 50 µm.

**Figure 5 antibiotics-12-01319-f005:**
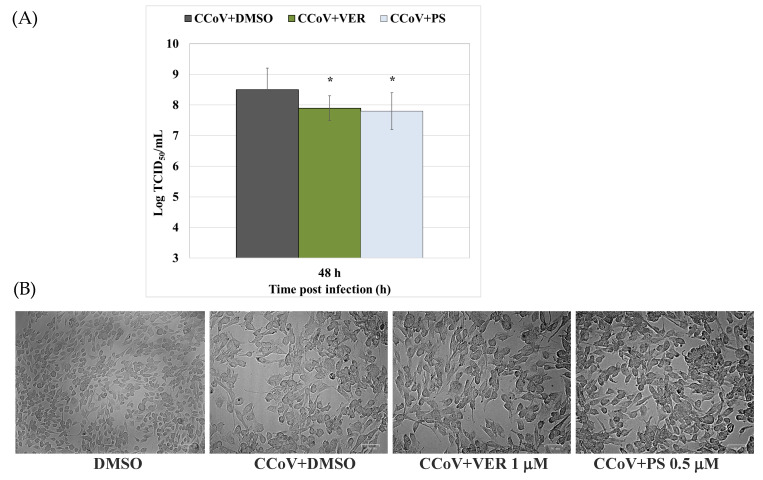
Funicone-like compounds reduce virus yield during CCoV infection. (**A**) For viral titer, cells were infected with CCoV and treated or not with VER and PS. After 48 h of infection, virus titer was evaluated by RT-qPCR. (**B**) Cytopathic effect (CPE), observed by light microscope (50 and 100 μm). Virus yield was evaluated by RT-qPCR using a standard curve created by amplifying serial known dilutions (three replicas/dilution) of the virus and plotting Log TCID_50_/mL against average Ct number. The comparison between CCoV-infected cells treated with funicone-like compounds vs. untreated infected cells was significant (* *p* < 0.05).

**Figure 6 antibiotics-12-01319-f006:**
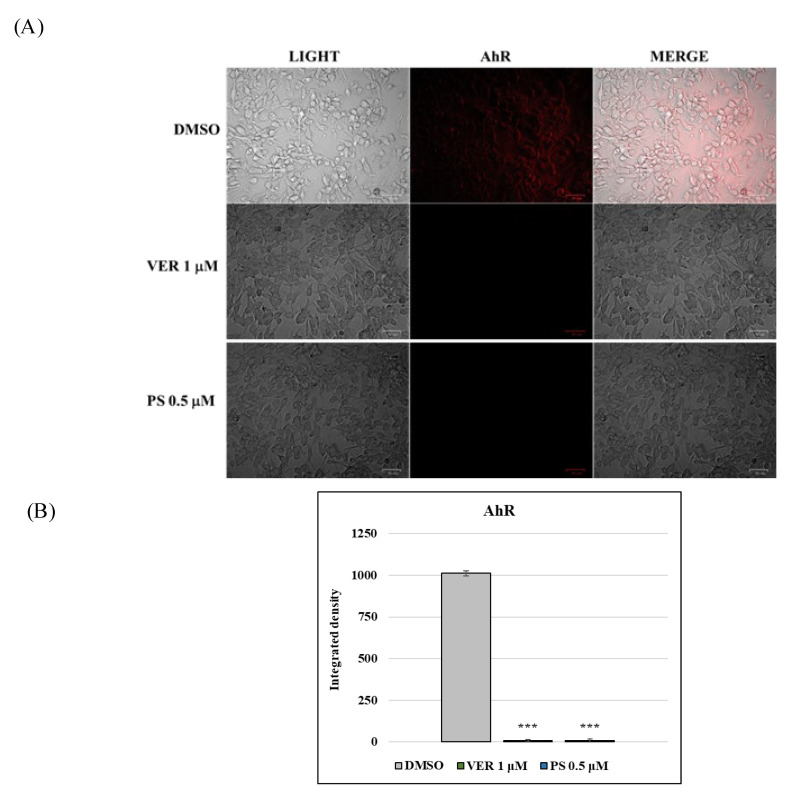
Funicone-like compounds reduce the expression of AhR. (**A**) DMSO control cells expressed AhR, which was significantly lessened by both VER and PS. Scale bar 50 µm. (**B**) Bars designate the mean ratio developed from the integrated density of AhR expression measured by ImageJ. Error bars correspond to standard deviation quantification and significant differences are indicated by probability *p*. *** *p* < 0.001.

**Figure 7 antibiotics-12-01319-f007:**
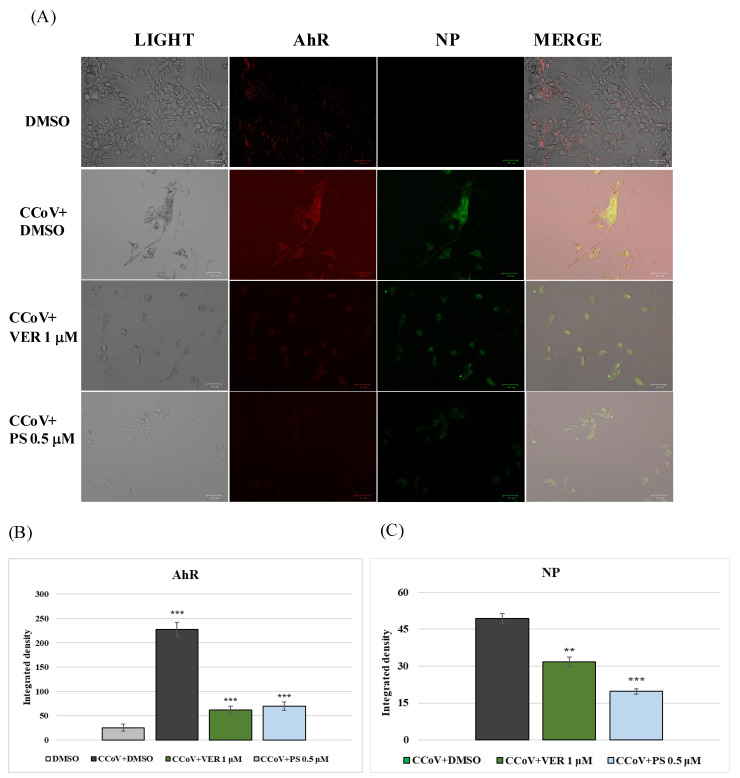
Funicone-like compounds reduce the expression of both AhR and viral nuclear protein (NP). CCoV-infected A72 cells were treated with VER and PS. After 24 h of infection, IF staining to detect AhR (red fluorescence) and NP (green fluorescence) was performed. (**A**) In both groups treated with funicone-like compounds, we observed a high reduction in the expression of AhR and NP. Scale bar 50 µm. Bars identify the mean ratio resulting from the integrated density of (**B**) AhR and (**C**) NP expression, analyzed by ImageJ. Error bars are the standard deviation quantification and significant differences are indicated by probability *p*. ** *p* < 0.01 and *** *p* < 0.001.

**Figure 8 antibiotics-12-01319-f008:**
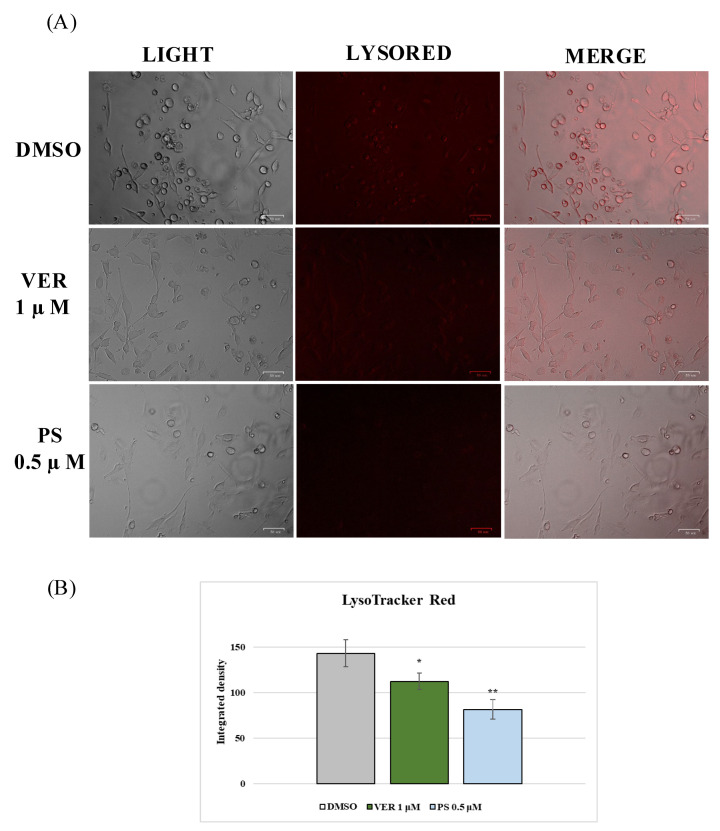
Funicone-like compounds deacidify lysosomes in A72 cells. (**A**) LysoRed staining of DMSO control group compared to cells treated with VER and PS. Scale bar 50 µm. (**B**) Bars designate the mean ratio obtained by the integrated density of LysoTracker calculated by ImageJ. Error bars represent standard deviation quantification and significant differences are indicated by probability *p*. * *p* < 0.05 and ** *p* < 0.01.

**Figure 9 antibiotics-12-01319-f009:**
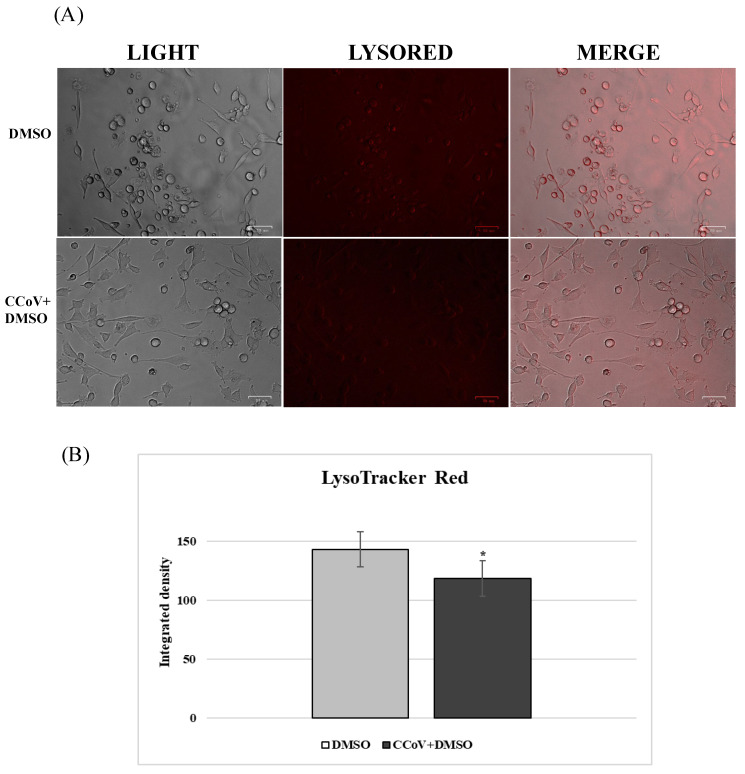
CCoV deacidifies lysosomes during infection. (**A**) LysoRed staining of CCoV-infected cells compared to Control group. Scale bar 50 µm. (**B**) Bars indicate the mean ratio obtained by the integrated density of LysoTracker measured by ImageJ. Error bars indicate standard deviation quantification and significant differences are indicated by probability *p*. * *p* < 0.05.

**Figure 10 antibiotics-12-01319-f010:**
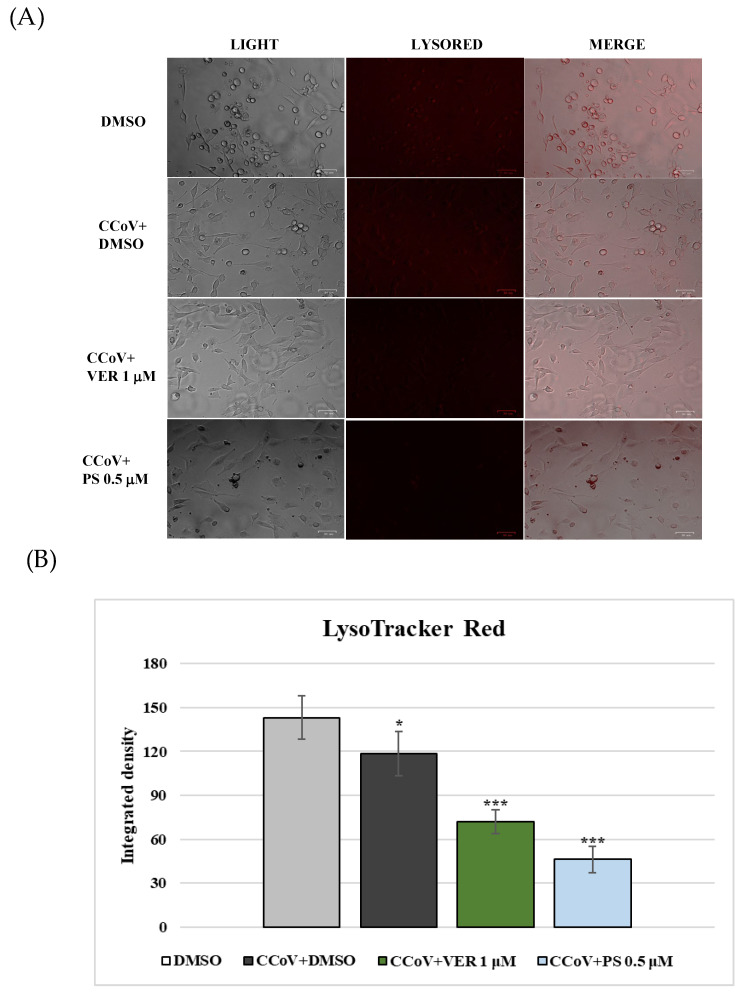
Funicone-like compounds deacidify lysosomes during CCoV infection (**A**) LysoRed staining of CCoV-infected cells compared to CCoV-infected cells treated with VER and PS. Scale bar 26 µm. (**B**) Bars indicate the mean ratio obtained by the integrated density of LysoTracker measured by ImageJ. Error bars indicate standard deviation quantification and significant differences are indicated by probability *p*. * *p* < 0.05 and *** *p* < 0.001.

## Data Availability

The data that support the findings of this study are available from the corresponding author upon reasonable request.

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
