# Peer review of "In Vitro Evaluation of Antiviral Activities of Funicone-like Compounds Vermistatin and Penisimplicissin against Canine Coronavirus Infection"

_antibiotics, 2023, doi:10.3390/antibiotics12081319_

Round 1

Reviewer 1 Report

The authors attempted to investigate the potential antiviral effects of VER and PS toward CCoV infection in 68 canine fibrosarcoma cells (A72).

This is an interesting study and the work done is commendable.

However, the presentation is not coherent, the methodology is not adequately explained and is also presented at the end of the study whereas it should have been after the introduction. This makes the study incoherent and cannot be adequately followed.

The authors should rearrange the study to give it more meaning.

The English is generally fine with some minor edits required

Author Response

The authors attempted to investigate the potential antiviral effects of VER and PS toward CCoV infection in 68 canine fibrosarcoma cells (A72).

This is an interesting study and the work done is commendable.

We wish to thank the referee for his/her positive comments.

However, the presentation is not coherent, the methodology is not adequately explained and is also presented at the end of the study whereas it should have been after the introduction. This makes the study incoherent and cannot be adequately followed.

The authors should rearrange the study to give it more meaning.

According to the “Instruction for Authors” of the journal “Antibiotics” (https://www.mdpi.com/journal/antibiotics/instructions), the manuscript was prepared using the Microsoft Word Template in which the Results section is placed right after the Introduction section.

As suggested, we modified the Material and Methods section in order to improve the methodology explanations.

Reviewer 2 Report

Authors have carried out an in vitro approach initially selecting two funicone-like compounds (vermistatin and penisimplicissin) and investigating their inhibitory activity towards CCoV infection by testing for their cytotoxicity and antiviral activity. Preliminary data show promising results for these compounds. This is a short and straightforward manuscript that is well written and provides novel information on potentially new therapeutic options. In my opinion it is likely to be well received by the scientific community and for all the above I advise publication after minor comments/concerns are addressed

Line 51 “and insecticidal”

In Results, Line 80, please delete “as reported in Material and Methods section,”. In fact, I would delete subsection 2.1 and place relevant information on Materials and Methods only

DMSO and compounds (and concentrations in the 4 pictures of figure 2 are of suboptimal quality. I would also increase the size of axis Y and X on C and D. Also no need to present colours for the bars subtitles. I would just leave the subtitles below each bar and try to separate them a bit more

Figure 3 has the same problems

On Figure 6, colour assignment for PES and VER seem to be of the same red. Please separate bars for clarity. Do the same for Figs 7-10.

On materials and methods, line 271, change to “Cultures of T. pinophilus (strain LT6) were prepared”

Lie 273. Whats is the name of the mixer and with which speed and for how long?

How did you calculate extraction efficiency with spiking of MNV? what was the initial concentration of MNV and what were the copie numbers detected after extraction? How did you calculate these?

Author Response

Authors have carried out an in vitro approach initially selecting two funicone-like compounds (vermistatin and penisimplicissin) and investigating their inhibitory activity towards CCoV infection by testing for their cytotoxicity and antiviral activity. Preliminary data show promising results for these compounds. This is a short and straightforward manuscript that is well written and provides novel information on potentially new therapeutic options. In my opinion it is likely to be well received by the scientific community and for all the above I advise publication after minor comments/concerns are addressed

 We wish to thank the referee for his/her positive comments and useful suggestions.

Line 51 “and insecticidal”

 Modified

In Results, Line 80, please delete “as reported in Material and Methods section,”. In fact, I would delete subsection 2.1 and place relevant information on Materials and Methods only

 As suggested, we delated “as reported in Material and Methods section”. Moreover, subsections 2.1 and 2.2 have been merged.

DMSO and compounds (and concentrations in the 4 pictures of figure 2 are of suboptimal quality. I would also increase the size of axis Y and X on C and D. Also no need to present colours for the bars subtitles. I would just leave the subtitles below each bar and try to separate them a bit more

 Figure 3 has the same problems

We changed Figures 2 and 3. Please, see the new Figures in the manuscript.

On Figure 6, colour assignment for PES and VER seem to be of the same red. Please separate bars for clarity. Do the same for Figs 7-10.

 As suggested, we modified the colors of the bars in figs 6-10.

On materials and methods, line 271, change to “Cultures of T. pinophilus (strain LT6) were prepared”

 Modified

Lie 273. Whats is the name of the mixer and with which speed and for how long?

 Added

How did you calculate extraction efficiency with spiking of MNV? what was the initial concentration of MNV and what were the copie numbers detected after extraction? How did you calculate these?

MNV was not use to calculate extraction efficiency but, as already stated in the paper, to evaluate presence of PCR inhibitors. More details about this procedure were reported in the paper (paragraph 4.8 lines 341-351) as kindly requested. Information on MNV and its initial concentration were also added.

Reviewer 3 Report

Manuscript entitled ‘Evaluation of Antiviral Activities of Funicone-like Compounds
Vermistatin and Penisimplicissin against Canine Coronavirus Infection’
follows the trend for constant searching for novel safe antiviral compounds. Due to the nature of viral infections this research direction remained valid for decades. The present research is a continuation of previous studies conducted by the authors. Although the manuscript is interesting and novel, there are some problems with a title and flaws in the procedures and presentation of the data.

1. Title

The manuscript title is somewhat misleading. The phrase ‘against canine coronavirus infection’ may imply animal research. The study was conducted under in vitro conditions only and the title should clearly inform readers about it.

2. Methodology

a) IC50 value calculation

Figure 2 title is: ‘Identification of IC50 of VER and PS at various doses and development of dose-response curve’

The IC50 value has not been identified by the authors and simply cannot be calculated based on the presented data. The authors dose-response curves are incomplete. To calculate IC50, you would need a series of dose-response data (not just three of them) ant the values of cell viability (in this very case) should be in the range of 0-100%. It is impossible to calculate the value of compound that reduces cell viability by 50% if all of the analysed values are bigger than 50%. Please read a simple guide:

https://www.graphpad.com/support/faq/50-of-what-how-exactly-are-ic50-and-ec50-defined/

https://www.graphpad.com/support/faq/how-to-determine-an-icsub50sub/

b) TCID50 value calculation

The authors stated: ‘virus titer was evaluated by RT-qPCR’.

This approach is very unusual. The TCID50 (50% Tissue Culture Infectious Dose) assay is the endpoint dilution assay that quantifies the amount of virus required to produce a cytopathic effect (CPE) in 50% of inoculated tissue culture cells. To calculate the TCID50 values viral CPE should be monitored daily under inverted microscope and recorded.

The RT-qPCR method is much more sophisticated tool and the results obtained are usually presented as viral RNA copy number per 1 µL of RNA or relative expression of viral RNA (in comparison to virus-infected untreated cells). What is more, the authors could use this method for both extracellular and intracellular viral RNA quantification.

There are correlations between TCID50/mL and Ct values generated by real-time PCR, but they should be described in great detail in the manuscript what has not been done by the authors.

During the study viral CPE was recorded by the authors. What kept the authors from calculating real TCID50 values?

3. Toxic concentrations of analysed compounds

As one could see from the Figure 1, two concentrations of the compounds used in the study, e.g 2.5 µM of VER and 1 µM of PS were toxic to cells and significantly decreased cell viability. The cytotoxicity testing in the in vitro studies is performed in order to determine the range of safe concentrations. Toxic concentrations should be excluded from further study, otherwise, the results obtained may be unreliable. Why did the authors include those concentrations in the next stage of the study (Fig. 3)?

4. Data presentation

a) fluorescent staining

Micrographs of LysoRed staining (Figures 8 and 9) are of very poor quality. Lysosomal staining results in rather bright cells, since lysosomes are usually distributed throughout the whole fibroblast. In Fig. 8 and 9 the background fluorescence is similar to the fluorescence of the cells. At least the control cells should be as bright as the cells in the Fig. 4 (stained with AO).

b) lack of control (mock infected cells)

In all the micrograph panels showing cells inoculated with virus (Figures 3, 4, 5, 7 and 10) there should be included micrographs of control (mock-infected) cells for comparison.

c) magnification problems

Figure 4: Why the micrographs showing various treatments were taken under different magnifications?

c) colours in graphs

The same colour of bars representing various treatments is misleading (Figures 6-10). Why not use the same colour palette as in Figures 2 and 3 or 5?

Author Response

Manuscript entitled ‘Evaluation of Antiviral Activities of Funicone-like Compounds
Vermistatin and Penisimplicissin against Canine Coronavirus Infection’ follows the trend for constant searching for novel safe antiviral compounds. Due to the nature of viral infections this research direction remained valid for decades. The present research is a continuation of previous studies conducted by the authors. Although the manuscript is interesting and novel, there are some problems with a title and flaws in the procedures and presentation of the data.

We wish to thank the referee for his/her useful suggestions

  1. Title

The manuscript title is somewhat misleading. The phrase ‘against canine coronavirus infection’ may imply animal research. The study was conducted under in vitro conditions only and the title should clearly inform readers about it.

As suggested, the title has been modified

  1. Methodology
  2. a) IC50value calculation

Figure 2 title is: ‘Identification of IC50 of VER and PS at various doses and development of dose-response curve’

The IC50 value has not been identified by the authors and simply cannot be calculated based on the presented data. The authors dose-response curves are incomplete. To calculate IC50, you would need a series of dose-response data (not just three of them) ant the values of cell viability (in this very case) should be in the range of 0-100%. It is impossible to calculate the value of compound that reduces cell viability by 50% if all of the analysed values are bigger than 50%. Please read a simple guide:

https://www.graphpad.com/support/faq/50-of-what-how-exactly-are-ic50-and-ec50-defined/

https://www.graphpad.com/support/faq/how-to-determine-an-icsub50sub/

As requested by the Reviewer, we calculated IC50, considering six different doses of both funicones, including concentrations which reduced cell viability by 50%. Please, see new Figure 2.

In A72 cells, the IC50 was obtained with 4.2556 μM VER and 4.9562 PS μM, respectively.

  1. b) TCID50value calculation

The authors stated: ‘virus titer was evaluated by RT-qPCR’.

This approach is very unusual. The TCID50 (50% Tissue Culture Infectious Dose) assay is the endpoint dilution assay that quantifies the amount of virus required to produce a cytopathic effect (CPE) in 50% of inoculated tissue culture cells. To calculate the TCID50 values viral CPE should be monitored daily under inverted microscope and recorded.

The RT-qPCR method is much more sophisticated tool and the results obtained are usually presented as viral RNA copy number per 1 µL of RNA or relative expression of viral RNA (in comparison to virus-infected untreated cells). What is more, the authors could use this method for both extracellular and intracellular viral RNA quantification.

There are correlations between TCID50/mL and Ct values generated by real-time PCR, but they should be described in great detail in the manuscript what has not been done by the authors.

During the study viral CPE was recorded by the authors. What kept the authors from calculating real TCID50 values?

Funicones ability to decrease virus yield was evaluated by RT-qPCR. Quantification was made by the mean of a standard curve as we have also done in other already published experimental researches (see references 18,19,31). The standard curve was constructed based on the average Ct values of three replicates against the Log of known amount of the virus (expressed in TCID50/mL). As requested, we better described the protocol in the paragraph 4.9 Real-Time RT-PCR for CCoV quantification. Moreover, as described in our funded research project IZS ME 05/21RC, we analyzed virus titer by RT-qPCR and not by TCID50.

  1. Toxic concentrations of analysed compounds

As one could see from the Figure 1, two concentrations of the compounds used in the study, e.g 2.5 µM of VER and 1 µM of PS were toxic to cells and significantly decreased cell viability. The cytotoxicity testing in the in vitro studies is performed in order to determine the range of safe concentrations. Toxic concentrations should be excluded from further study, otherwise, the results obtained may be unreliable. Why did the authors include those concentrations in the next stage of the study (Fig. 3)?

Please accept our apologies for the mistake. We cut them. Please, see the new Fig. 3.

  1. Data presentation
  2. a) fluorescent staining

Micrographs of LysoRed staining (Figures 8 and 9) are of very poor quality. Lysosomal staining results in rather bright cells, since lysosomes are usually distributed throughout the whole fibroblast. In Fig. 8 and 9 the background fluorescence is similar to the fluorescence of the cells. At least the control cells should be as bright as the cells in the Fig. 4 (stained with AO).

We improved the quality of them. Please, see new figures 8 and 9.

  1. b) lack of control (mock infected cells)

In all the micrograph panels showing cells inoculated with virus (Figures 3, 4, 5, 7 and 10) there should be included micrographs of control (mock-infected) cells for comparison.

  1. c) magnification problems

Figure 4: Why the micrographs showing various treatments were taken under different magnifications?

All the micrographs were taken using the same magnifications, at different magnification. For PS groups, we have chosen a magnification that shows as no signs of cell death were detected in a larger area (50 μm) and a magnification 100 μm for DMSO

  1. c) colours in graphs

The same colour of bars representing various treatments is misleading (Figures 6-10). Why not use the same colour palette as in Figures 2 and 3 or 5?

We changed them.

Round 2

Reviewer 3 Report

The revised manuscript is acceptable for publication.